# The Impact of Older Age on Functional Recovery and Quality of Life Outcomes after Surgical Decompression for Degenerative Cervical Myelopathy: Results from an Ambispective, Propensity-Matched Analysis from the CSM-NA and CSM-I International, Multi-Center Studies

**DOI:** 10.3390/jcm8101708

**Published:** 2019-10-17

**Authors:** Jamie R. F. Wilson, Jetan H. Badhiwala, Fan Jiang, Jefferson R. Wilson, Branko Kopjar, Alexander R. Vaccaro, Michael G. Fehlings

**Affiliations:** 1Division of Neurosurgery, Department of Surgery, University of Toronto, Toronto, ON M5T2S8, Canada; jamie.wilson@mail.utoronto.ca (J.R.F.W.); jetan.badhiwala@medportal.ca (J.H.B.); frank.jiang@mail.utoronto.ca (F.J.); WilsonJeff@smh.ca (J.R.W.); 2Division of Neurosurgery, Toronto Western Hospital, University Health Network, Toronto, ON M5T2S8, Canada; 3Division of Neurosurgery, St. Michael’s Hospital, University Health Network, Toronto, ON M5B1W8, Canada; 4Department of Health Services, School of Public Health, University of Washington, Seattle, WA 98195, USA; branko.kopjar@nor-consult.com; 5Department of Orthopedic Surgery, The Rothman Institute at Thomas Jefferson University, Philadelphia, PA 19107, USA; alex.vaccaro@rothmaninstitute.com

**Keywords:** degenerative cervical myelopathy, elderly, old age, outcomes, complications, mJOA, SF-36

## Abstract

Background: The effect on functional and quality of life (QOL) outcomes of surgery in elderly degenerative cervical myelopathy (DCM) patients has not been definitively established. Objective: To evaluate the effect of older age on the functional and QOL outcomes after surgery in an international, multi-center cohort of patients with DCM. Methods: 107 patients aged over 70 years old (mean 75.6 ± 4.4 years) were enrolled in the AOSpine CSM-North America and International studies. A propensity-matched cohort of 107 patients was generated from the remaining 650 adults aged <70 years old (mean 56.3 ± 9.6 years), matched to gender, complexity of surgery, co-morbidities, and baseline functional impairment (modified Japanese Orthopedic Association scale (mJOA). Functional, disability, and QOL outcomes were compared at baseline and at two years post-operatively, along with peri-operative adverse events. Results: Both cohorts were equivalently matched. At two years, both cohorts showed significant functional improvement from the baseline but the magnitude was greater in the younger cohort (mJOA 3.8 (3.2–4.4) vs. 2.6 (2.0–3.3); *p* = 0.007). This difference between groups was also observed in the SF-36 physical component summary (PCS) and mental component summary (MCS) outcomes (*p* = <0.001, *p* = 0.007), but not present in the neck disability index (NDI) scores (*p* = 0.094). Adverse events were non-significantly higher in the elderly cohort (22.4% vs. 15%; *p* = 0.161). Conclusions: Elderly patients showed an improvement in functional and QOL outcomes after surgery for DCM, but the magnitude of improvement was less when compared to the matched younger adult cohort. An age over 70 was not associated with an increased risk of adverse events.

## 1. Introduction

Degenerative cervical myelopathy (DCM) is a family of non-traumatic spinal cord injuries that contribute to spinal cord compression and the progressive onset of neurological deficits [1,2]. DCM encompasses a number of related conditions such as cervical spondylotic myelopathy (CSM), ossification of the posterior longitudinal ligament (OPLL), ossification of the ligamentum flavum, degenerative disc disease, and various congenital malformations that cause stenosis or instability leading to eventual spinal cord dysfunction [2]. It is now the commonest form of spinal cord dysfunction in adults [3], and has become an increasingly important area of focus for spine surgeons and clinician scientists in recent years [4,5,6,7,8,9]. This renewed focus has primarily been driven by a greater understanding of the benefit of timely intervention [5,7,10,11], and international clinical practice guidelines for the management of DCM were published in 2017 [7].

The aging population, juxtaposed with developments in medical technologies, has led to the “epidemiological transition”; the shift away from traditional causes of disease and mortality (infectious, nutritional deficiency, parasitic) toward chronic and degenerative diseases [3]. The incidence of DCM increases with age [3]; cervical spondylosis affects half of individuals aged 50–59 years, and nearly all individuals over the age of 70 [12]. The natural history of DCM is one of progressive neurological deficit, with the potential irreversible loss of dexterity, quadriplegia, and sphincter dysfunction [3,12,13]. DCM is set to become a major cause of chronic disability and global disease burden, if the current demographic expansion continues [14].

Decompressive surgery has been shown to not only halt neurological deterioration in DCM, but can produce a significant recovery in the neurological impairment that individuals experience [7,15]. The extent of functional and quality of life (QOL) improvement after surgery is influenced by the duration of symptoms, severity of functional impairment at presentation, the presence of co-morbidities, tobacco smoking status as well as age at presentation [8,11,16,17,18,19,20]. Elderly patients often present with a longer duration of symptoms, and more severe functional deficit compared to younger adults with DCM [11,12,18,19]. In addition, elderly patients have a higher prevalence of degenerative conditions, co-morbidities, an increased risk of osteoporosis, and higher peri-operative morbidity and mortality when compared to younger patients [3,12,21].

Age is a common surrogate for frailty or physiological reserve. The effect that frailty alone has on the effect of spine surgery has been studied through various means, and seems most relevant in the practice of spinal deformity where the procedures are often invasive and of long duration [22,23,24,25,26,27,28]. However, the significance of age on the effect of surgical outcomes for DCM is unclear. Available evidence would suggest that increasing age is associated with a worse functional outcome after surgical decompression [29,30,31,32]. However, many articles based on retrospective case series have also presented evidence demonstrating no significant differences in outcomes in terms of the modified Japanese Orthopedic Association score (mJOA), Nurick, and SF-36 scores when directly compared to standardized, younger patient cohorts [33,34,35]. Older patients are also more likely to undergo posterior surgery with a higher number of operated cervical levels when compared to younger adults [15,16,19].

The objectives of the current study were to (1) explore the functional and QOL impairment of adults over 70 years old with DCM compared to younger adults; (2) define how the variable of age over 70 alone can contribute to functional outcomes after DCM surgery, when accounting for other common age-related variables; and (3) define how elderly age can affect the QOL outcomes when compared to an equivalent adult cohort. We hypothesized that (1) adults over 70 (when compared to a matched cohort of younger adults with the same functional impairment) would exhibit less quality of life improvement; (2) when the effects of age-related co-morbidities and surgical factors were adjusted for, age over 70 would remain a significant risk factor for worse functional improvement after surgery; and (3) younger adults would have a more sustained improvement in QOL outcomes when compared to the elderly cohort.

## 2. Methods

### 2.1. Subjects

The AOSpine CSM-NA study recruited 278 patients with symptomatic DCM and correlating magnetic resonance imaging findings over a duration of two years from 12 North American centers [15]. The AOSpine CSM-International study recruited 479 symptomatic DCM patients from six Asian, five European, three Latin American, and two North American sites over a duration of four years [17]. Both studies were prospective observational studies, and patients were eligible for enrolment if they met the following inclusion criteria: (1) symptomatic DCM with one or more clinical signs of myelopathy; (2) imaging evidence compatible with spinal cord compression; (3) aged 18 years or older; and (4) no previous history of cervical spine procedures. Exclusion criteria included patients with malignancy or metastatic disease, active systemic infection, trauma, rheumatoid arthritis (or other inflammatory disease such as ankylosing spondylitis), symptomatic tandem lumbar stenosis, or asymptomatic DCM patients. All centers obtained approval from their respective local ethical boards prior to commencement of the study.

### 2.2. Baseline Characteristics

All patients enrolled in both studies had demographic data recorded prior to surgery (age, race, socioeconomic status, tobacco smoking status, body mass index (BMI), presence of co-morbidities) together with a focused myelopathy history including duration of symptoms, clinical signs, and etiology of DCM. All patients underwent surgical treatment, with the surgical approach and number of spinal levels operated left to the discretion of the treating surgeon. Peri-operative demographics were recorded including the surgical procedure of choice (anterior discectomy, corpectomy, laminoplasty, laminectomy with or without instrumented fusion). Post-operative complications up to 24 months after surgery were recorded.

### 2.3. Outcomes

Functional and QOL assessments were performed prior to surgery at baseline, then six, 12, and 24 months after surgery. Functional status was assessed by the mJOA scale; a standardized assessment of neurological and functional impairment that was administered by investigators [4,5,36]. Disability and QOL assessments were self-reported outcome measures in the form of the Neck Disability Index (NDI—specific to cervical degenerative pathologies) and the Short Form-36 version 2 (SF-36—a generic health-related QOL measurement). The SF-36 was further separated into the SF-36 Physical Component Summary (PCS) and Mental Component Summary (MCS) in an effort to distinguish between the patient-reported perception of physical health compared to mental and emotional well-being.

### 2.4. Statistical Analysis

All statistical analyses were performed using Stata 15 (Stata Corp, College Station, TX, USA) and R version 3.5.1 (R Foundation for Statistical Computing, Vienna, Austria) with an a priori specified significance level of *p* = 0.05 (two-tailed). Descriptive statistics are listed as mean and standard deviation (SD) for continuous variables and count and percentage for categorical variables.

#### 2.4.1. Propensity Score Matching

The propensity score matching algorithm was developed and executed using the ‘MatchIt’ package for R statistical software by one of the coauthors (J.H.B.). Variables to be included as covariates in the generation of propensity scores were defined a priori by author consensus of clinical relevance.

Propensity scores were calculated as the probability of age ≥ 70 years versus age < 70 years using the logit method with the baseline mJOA score, duration of DCM symptoms, cardiac disease, diabetes, smoking (current), psychiatric disease, surgical approach, and number of operated levels as covariates (independent variables). Propensity score matching was performed in a one-to-one ratio using the ‘optimal matching’ technique to minimize the average absolute distance across all matched pairs. This resulted in two study groups (age < 70 vs. age ≥ 70) adjusted for the baseline covariates specified above. The baseline characteristics were compared between the study groups by the t-test for continuous variables and chi-square test for categorical variables.

#### 2.4.2. Analysis of Outcomes

Two year outcomes for functional status (mJOA score), disability (NDI), and health-related quality of life (HRQOL; SF-36 PCS, and MCS) were compared between the study groups using an analysis of covariance (ANCOVA) adjusting for baseline score. Effect sizes for each outcome measure were summarized by β coefficients (mean difference) and associated 95% confidence intervals (CIs).

## 3. Results

Both young and older cohorts were matched sufficiently in terms of sex, duration of symptoms, smoking history, co-morbidities, number of levels, surgical approach, and baseline functional impairment (Table 1). The mean age of the younger cohort was 56.3 ± 9.6 when compared to 75.6 ± 4.4 in the older cohort (*p* < 0.001). The younger cohort demonstrated a significantly worse baseline SF-36 Physical Component Score (PCS, 30.7 ± 8.2 vs. 33.5 ± 8.8 (*p* = 0.019)) and Mental Component Score (MCS, 38.4 ± 12.8 vs. 43.0 ± 13.1, (*p* = 0.011)) when compared to the older cohort.

Both cohorts demonstrated an improvement in functional impairment at two years, as defined by the mJOA scale (*p* = 0.001, see Table 2). Significant improvements were also seen in the NDI, PCS, and MCS scores, with the exception of the elderly MCS score (*p* = 0.077). The functional outcomes in the younger cohort were of a greater magnitude when compared to the older group (mean difference 3.8 ± 3.0 versus 2.6 ± 3.3; *p* = 0.007; see Table 3). This difference was also present in the QOL measurements (PCS and MCS, *p* < 0.001, *p* = 0.007). The change in NDI scores between the groups at two years showed no significant difference (*p* = 0.094).

The total number of all-cause adverse events over two years, including peri-operative complications and worsening functional impairment, was lower in the younger cohort (*n* = 16; 15%) when compared to the older cohort (*n* = 24; 22.4%; *p* = 0.161), but this was not statistically significant (see Table 4). Post-operative infection and hardware failure were equivalent between the younger and older cohorts (*p* = 0.701 and *p* = 0.313, respectively). The incidence of post-operative dysphagia was significantly higher in the older cohort when compared to the younger cohort (*n* = 6 compared to *n* = 0; *p* = 0.013). The incidence of the worsening of myelopathy symptoms was higher in the older cohort when compared to the younger cohort, which was just shy of statistical significance (*n* = 17 (15.9%) vs. *n* = 8 (7.5%); *p* = 0.055).

## 4. Discussion

Renewed focus on functional and quality of life outcomes after surgery for DCM in adults has come from rigorous scrutiny of the results of several large-scale studies in the last decade [2,4,5,7,10,17,37,38,39,40,41,42]. The traditional doctrine of the use of surgery to ‘arrest clinical progression’ has been replaced with a more modern approach with decompressive surgery used as a means to provide both functional and quality of life improvement [5,7]. The assessment of quality of life metrics has become particularly important in the examination of patients with mild functional impairment (or asymptomatic cord compression), and is the focus of a number of studies [5,7,43]. Careful evaluation has discovered a number of predictors of clinical outcomes from surgery for DCM, including baseline severity of impairment, duration of symptoms, presence of systemic or psychological co-morbidities, obesity, and of course age [11,18,21]. With simple univariate analysis of outcome measures, many studies have reported worse functional and QOL outcomes with increasing age after surgery for DCM [29,30,31,32]. However, elderly patients often present with increased duration and worse severity of symptoms, with an increased incidence of co-morbidities [11,18,29]. Many elderly patients also present a high, sometimes unacceptable, surgical risk profile that makes decisions regarding surgical management less appropriate. DCM is a disease that can cause a significant amount of potentially reversible or preventable neurological disability in the elderly population, and therefore concerted efforts should be made to improve decision-making strategies for this patient group.

The results of the current study demonstrate a number of important conclusions, some of which have not been described in previous literature. Firstly, even when matched for co-morbidities, duration of symptoms, sex, smoking status and severity of functional impairment, younger adults report significantly worse effects on their baseline physical and mental quality of life scores compared to the older aged cohort. This could be explained through 2 potential mechanisms. Firstly, as the younger cohort were of working age, the psychological impact of DCM may be greater if employment security is perceived to be threatened. Secondly, the elderly patients demonstrate psychological resilience or adaptations/support that mean the symptoms of DCM have a less pronounced effect on their quality of life. This is an interesting concept, and is an important consideration for clinicians managing elderly DCM patients.

All patients showed improved functional impairment scores after 2 years (see Table 2). Taking the older cohort in isolation, this is good evidence that surgery is an effective modality to produce functional improvements in elderly DCM patients. This is an important stand-alone conclusion from this study. When the change from baseline scores at 2 years is calculated, it becomes evident that the functional improvement seen after surgery is of a greater magnitude in younger patients compared to the older cohort. This conclusion is similar to those described from previous univariate analyses in prior studies [11,18,29,32,44]. Suggested mechanisms for this discrepancy include the fact that elderly patients often present with increased duration of symptoms at diagnosis, often have difficulty accessing specialist assessment and imaging, and also have less neurological plasticity or reserve compared to younger adults [11,12,18,45,46]. The level of disability as measured by the NDI was improved significantly by surgery for all patients, but the order of magnitude was similar for both age groups (*p* = 0.094).

All patients demonstrated improved SF-36 PCS and MCS from baseline at the 2-year interval, with the exception of the MCS scores in the elderly group. Both PCS and MCS metrics echoed the functional improvement in that the younger cohort had a significantly greater degree of improvement from baseline compared to the older cohort (*p* < 0.001, *p* = 0.007). This provides good evidence that surgery leads to increased physical and mental perception of quality of life outcomes in all patients undergoing surgery for DCM. It appears that older patients’ mental perception of quality of life remains stable throughout the treatment period, despite having a worse functional improvement, and less magnitude of quality of life improvement overall. This study is the first to report this difference in quality of life measures between younger and older aged cohorts and has potentially large implications on pre-surgical assessment and counselling in older patients with DCM. However, age-related effects on health perception have been found to influence the SF-36 score, and the contribution from the perception of physical effects often misses a number of important determinants of overall quality of life in older persons [47,48]. For these reasons, it has been suggested that the focus of quality of life determinants in the elderly should be weighted toward mental components rather than the physical [48]. This also raises certain questions about the efficacy of functional assessment measures for DCM in the elderly, but this was beyond the scope of the current study.

All forms of adverse events over two years were higher in the elderly cohort when compared to the younger adults, but not significantly so (*p* = 0.161, see Table 4). The risk of infection, hardware failure, CSF leak, and need for revision surgery were all equivalent between groups. The incidence of dysphagia after the anterior approach was significantly greater in the elderly cohort, which has been well described [49]. The incidence of worsening functional impairment after surgery was non-significantly higher in the elderly cohort, which is also consistent with previously described studies and may be a reflection of the disease process in DCM, rather than directly related to complications from surgery [11,40,50]. These findings suggest that, contrary to popular opinion, surgery for DCM in the elderly does not carry a significantly higher risk of adverse events when compared to younger adults that are matched for co-morbidities and complexity of surgery. This is good evidence to demonstrate that age alone does not necessarily confer an increased risk of adverse events in DCM surgery, but that the other risk factors associated with older age (increased number of levels of pathology, posterior surgery, increased co-morbidities, etc.) ostensibly play more of a role in the determinant of peri-operative risk. This concept of frailty, and its association with the assessment of surgical risk, is emerging as an important tool for surgical decision making and has been a recent focus of interest in pathologies such as spinal cord injury and adult deformity surgery [51,52,53]. However, the impact of frailty on the outcomes from DCM appears to be less well defined.

There are limitations of the current study. The results were from the pooled analysis of two harmonized datasets, and although the data came from prospective, multi-center sources, the original studies were not designed or powered to measure the effect of old age on the outcomes after surgery for DCM. Although both cohorts were well matched (Table 1), there could exist significant heterogeneity between the groups or hidden confounders such as drug histories that may have affected the results. The use of propensity-matching methodology helps to reduce the over-fitting seen in mixed effects or regression models, but does have obvious effects in the sample size. Therefore, some aspects of the results that show trends and not significant differences may indicate that the results are underpowered in some areas.

## 5. Conclusions

To consider our previous hypotheses, elderly patients, when compared to younger adults (matched to functional impairment and age-related risk factors), exhibited better SF-36 PCS and MCS prior to surgery for DCM. All patients in this study demonstrated improved functional impairment two years after surgery, but the magnitude of improvement seen was greater in the younger cohort, even when baseline functional impairment and age-related risk factors were adjusted for. Elderly patients also showed improved QOL physical and mental component scores after surgery, but the extent of the increase in the physical component was reduced when compared to younger adults. Aside from the incidence of post-operative dysphagia, older age alone was not associated with a higher incidence of adverse events.

The authors believe that the results of this study provide good evidence that surgery for DCM in the elderly is effective in terms of both functional and QOL outcomes. Perhaps most importantly, these results demonstrate that the elderly DCM age group should have different expectations with regard to the extent of functional and QOL outcomes after surgery. Patients over the age of 70 with a diagnosis of DCM are likely to require specialist considerations and should be counseled appropriately with adjusted expectations. The exact degree to which each modifiable risk factor contributes to perioperative risk, and the components that affect functional and QOL outcomes, remain to be determined and should be an important focus of further research into the effects of aging on surgery for DCM. Developing a prediction model using age (or measures of frailty) and related covariate adjustment would significantly improve the calculation of the risk profile in DCM patients undergoing assessment for decompressive surgery.

## Figures and Tables

**Table 1 jcm-08-01708-t001:** Baseline demographics of the elderly cohort and younger cohort after propensity matching for gender, duration of symptoms, smoking status, co-morbidities, surgical factors, and baseline functional impairment.

Variable	Age < 70, *N* = 107	Age ≥ 70, *N* = 107	*p* Value
Demographics			
Age (years)—mean ± SD	56.3 ± 9.6	75.6 ± 4.4	<0.001 *
Female gender—No. (%)	68 (63.6)	68 (63.6)	1.000
Duration of symptoms—No. (%)	32.1 ± 49.7	26.9 ± 37.1	0.388
Smoker—No. (%)	6 (5.6)	10 (9.3)	0.299
Diabetes—No. (%)	14 (13.1)	15 (14.0)	0.842
Cardiovascular disease—No. (%)	70 (65.4)	74 (69.2)	0.560
Pulmonary disease—No. (%)	10 (9.3)	10 (9.3)	1.000
Psychiatric disease—No. (%)	8 (7.5)	8 (7.5)	1.000
Surgical factors			
Number of levels	3.2 ± 1.3	3.3 ± 1.3	0.591
Approach			0.889
Anterior	36 (33.6)	34 (31.8)	
Posterior	66 (61.7)	69 (64.5)	
Combined	5 (4.7)	4 (3.7)	
Functional status, disability, and QOL			
mJOA score	10.9 ± 2.8	11.0 ± 2.7	0.902
NDI	43.4 ± 20.5	39.5 ± 19.5	0.164
SF-36 PCS	30.7 ± 8.2	33.5 ± 8.8	0.019 *
SF-36 MCS	38.4 ± 12.8	43.0 ± 13.1	0.011 *

mJOA = modified Japanese Orthopedic Association scale, NDI = Neck Disability Index, SF-36 PCS = SF-36 Physical Component Score, SF-36 MCS = SF-36 Mental Component Score. * *p* value < 0.05.

**Table 2 jcm-08-01708-t002:** Functional, disability, and quality of life assessment scores of both cohorts at the two-year interval.

	Baseline	2 Years	MD (95% CI)	*p* Value
Age < 70 years				
mJOA	10.9 ± 2.8	14.8 ± 2.5	3.8 (3.2 to 4.4)	<0.001
NDI	43.4 ± 20.5	27.4 ± 18.6	−16.1 (−19.7 to −12.5)	<0.001
SF36-PCS	30.7 ± 8.2	38.9 ± 10.8	8.3 (6.5 to 10.2)	<0.001
SF-36 MCS	38.4 ± 12.8	46.8 ± 13.0	8.0 (5.3 to 10.8)	<0.001
Age ≥ 70 years				
mJOA	11.0 ± 2.7	13.6 ± 2.9	2.6 (2.0 to 3.3)	<0.001
NDI	39.5 ± 19.5	28.2 ± 17.8	−11.4 (−15.6 to −7.2)	<0.001
SF36-PCS	33.5 ± 8.8	36.9 ± 10.0	3.4 (1.4 to 5.5)	0.001
SF-36 MCS	43.0 ± 13.1	45.7 ± 13.7	2.6 (−0.3 to 5.5)	0.077

mJOA = modified Japanese Orthopedic Association scale, NDI = Neck Disability Index, SF-36 PCS = SF-36 Physical Component Score, SF-36 MCS = SF-36 Mental Component Score, MD = Mean Difference.

**Table 3 jcm-08-01708-t003:** Delta values (change in the scores from baseline) at two years for younger and older cohorts for all outcomes.

Outcome	Age < 70 Years	Age ≥ 70 Years	MD (95% CI)	*p* Value
ΔmJOA	3.8 ± 3.0	2.6 ± 3.3	1.2 (0.3 to 2.1)	0.007
ΔNDI	−16.1 ± 18.4	−11.4 ± 20.9	−4.7 (−10.2 to 0.8)	0.094
ΔPCS	8.3 ± 9.5	3.4 ± 10.6	4.9 (2.2 to 7.7)	<0.001
ΔMCS	8.0 ± 14.2	2.6 ± 14.9	5.5 (1.5 to 9.4)	0.007

mJOA = modified Japanese Orthopedic Association scale, NDI = Neck Disability Index, SF-36 PCS = SF-36 Physical Component Score, SF-36 MCS = SF-36 Mental Component Score, MD = Mean Difference.

**Table 4 jcm-08-01708-t004:** List of complications of elderly and younger adult cohorts over the two-year follow up.

Complication	Age < 70, *N* = 107	Age ≥ 70, *N* = 107	*p* Value
Infection	4 (3.7)	3 (2.8)	0.701
Deficit	2 (1.9)	0 (0)	0.155
CSF leak	2 (1.9)	1 (0.9)	0.561
Deformity	0 (0)	0 (0)	1.000
Hardware	3 (2.8)	1 (0.9)	0.313
Dysphagia	0 (0)	6 (5.6)	0.013 *
Dysphonia	0 (0)	1 (0.9)	0.316
Revision	0 (0)	1 (0.9)	0.316
Hematoma	0 (0)	1 (0.9)	0.316
Adjacent segment disease	1 (0.9)	0 (0)	0.316
Vascular injury	0 (0)	0 (0)	1.000
Worsening of myelopathy	8 (7.5)	17 (15.9)	0.055
Any complication	16 (15.0)	24 (22.4)	0.161

* *p* value < 0.05.

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
