# Peer review of "The Impact of Older Age on Functional Recovery and Quality of Life Outcomes after Surgical Decompression for Degenerative Cervical Myelopathy: Results from an Ambispective, Propensity-Matched Analysis from the CSM-NA and CSM-I International, Multi-Center Studies"

_jcm, 2019, doi:10.3390/jcm8101708_

Round 1

Reviewer 1 Report

Dear Authors

Interesting  article on an important issue. Maybe on aspect has to be added in the discussion that not age Rather the preoperative morbidity is more important 

Author Response

Reviewer 1:

Interesting  article on an important issue. Maybe on aspect has to be added in the discussion that not age Rather the preoperative morbidity is more important

Response:

Thank you. We have added a section on the concept of frailty and its application to spine surgery.

Reviewer 2 Report

I found this paper very impressive, clearly presented, interesting conclusions. Congratulations.

The work is valuable because it gives answers to questions that have not been resolved until now. As far as I know the topic seems to be quite original. I have no objections to both the language or the structure of the text and references. The conclusions are consistent with the results obtained.

Author Response

Reviewer 2:

I found this paper very impressive, clearly presented, interesting conclusions. Congratulations.

The work is valuable because it gives answers to questions that have not been resolved until now. As far as I know the topic seems to be quite original. I have no objections to both the language or the structure of the text and references. The conclusions are consistent with the results obtained.

Response:

Thank you for your comments.

Reviewer 3 Report

The manuscript treats a highly relevant clinical topic. Comparing the functional and QOL outcome between younger and older cohorts is challenging, though the propensity score matching method was well chosen and is appropriate in this case. It would be interesting for the readers to know how the matching process was performed in more detail (Who did it? One author or a group of authors after discussion? was it a blinded process? what did the authors do to minimise any bias in this process?).

The results are well presented. The manuscript is excellently discussed, the limitations have been correctly identified and the drawn conclusions seem appropriate.

Author Response

Reviewer 3:

The manuscript treats a highly relevant clinical topic. Comparing the functional and QOL outcome between younger and older cohorts is challenging, though the propensity score matching method was well chosen and is appropriate in this case. It would be interesting for the readers to know how the matching process was performed in more detail (Who did it? One author or a group of authors after discussion? was it a blinded process? what did the authors do to minimise any bias in this process?).

The results are well presented. The manuscript is excellently discussed, the limitations have been correctly identified and the drawn conclusions seem appropriate.

Response:

We have added a paragraph clarifying the propensity matching process, which was performed with statistical software.

Round 2

Reviewer 3 Report

no further comment.

Congratulations.

Author Response

No further changes were requested.